# VEnhancer: Generative Space-Time Enhancement for Video Generation

## Abstract

We present *VEnhancer*, a generative space-time enhancement method that can improve the existing AI-generated videos spatially and temporally through one video diffusion model. Given a generated low-quality video, our approach can increase its spatial and temporal resolution simultaneously with arbitrary up-sampling space and time scales by adding more details in spatial domain and synthesize detailed motion in temporal domain. Furthermore, VEnhancer is able to remove generated spatial artifacts and temporal flickering of generated videos. To achieve this, basing on a pretrained generative video prior, we train a **S**pace-**T**ime Controller and inject it to the prior as a condition on low-frame-rate and low-resolution videos. To effectively train this ST-Controller, we design *space-time data augmentation* to create diversified video training pairs as well as *video-aware conditioning* for realizing different augmentation parameters in both spatial and temporal dimensions. Benefiting from the above designs, VEnhancer can be end-to-end trained to enable multi-function in one single model. Extensive experiments show that VEnhancer surpasses existing state-of-the-art video super-resolution and space-time super-resolution methods in enhancing AI-generated videos. Moreover, VEnhancer is able to greatly improve the performance of open-source state-of-the-art text-to-video methods on video generation benchmark, VBench.

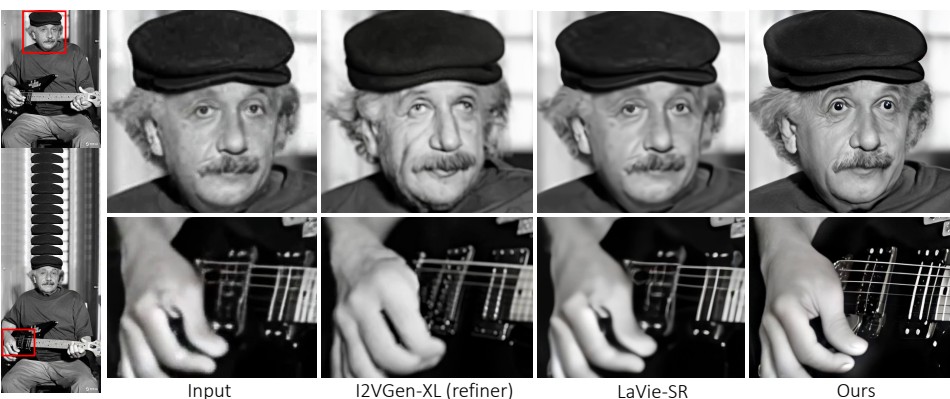

Figure 1: The enhanced screenshots for AI-generated videos (from Kling). **Prompt**: *Einstein plays guitar*. I2VGen-XL (refiner) (Zhang et al., 2023b) has successfully refined the video by removing distortions/artifacts, but suffers from severe identity change (*e.g.,* facial attributes) and blurry results. LaVie-SR (Wang et al., 2023b) could produce high-fidelity results but lacks generative ability in modifying and regenerating video content (*e.g.,* correcting the guitar strings). In contrast, our method could achieve effective refinement and output high-resolution videos with realistic texture details and good identity preservation. **Zoom in for best view.**

## 1 Introduction

With the advances of text-to-image generation (Rombach et al., 2022; Podell et al., 2023; Chen et al., 2023b; Gao et al., 2024) and large-scale video datasets with text description (Bain et al., 2021), there is fast development of text-to-video generative models (Guo et al., 2023; Chen et al., 2023c; Ho

et al., 2022; Blattmann et al., 2023b; Wang et al., 2023b;a; Chen et al., 2023a; 2024a; Gupta et al., 2023; Wu et al., 2023b). These developments enable users to generate compelling videos through textual descriptions of the desired content. One common solution (Ho et al., 2022; Blattmann et al., 2023b; Wang et al., 2023b; Blattmann et al., 2023a; Gupta et al., 2023) to obtain high-quality videos is to adopt cascaded pipelines, which stacks several video diffusion models, including text-to-video, temporal super-resolution and spatial super-resolution (S-SR) diffusion. These pipeline significantly reduce computation cost when generating high-resolution and high-frame-rate videos, but also pose several issues. First, using two different models for spatial and temporal enhancement separately is redundant, as they are strongly correlated. Also, the proposed diffusion-based spatial or temporal super-resolution (Blattmann et al., 2023b; Ho et al., 2022; Blattmann et al., 2023a; Wang et al., 2023b; Lin et al., 2024) have a limited flexibility, as they can only handle fixed interpolation ratio (*i.e.*, predicting three frames between two consecutive frames) or fixed upscaling factors (*i.e.*, $4\times$). Second, directly training diffusion models on synthesized video pairs may limit its generalization ability, as these models tailored on classic super-resolution task hallucinate high-frequency details, but cannot semantically improve the visual quality of input videos, such as eliminating distortions, artifacts, or recreating new contents (see LaVie-SR's results in Figure 1).

Another common approach is training another diffusion model to remove artifacts and to refine distorted content in the generated videos (Henschel et al., 2024; Zhang et al., 2023b). One example is I2VGen-XL (Zhang et al., 2023b), which follows the idea of image generation model SDXL (Podell et al., 2023)– it first upscales videos to higher resolutions using bilinear interpolation, and trains another diffusion refinement model, which will be used for video regeneration through a noising-denoising process (Meng et al., 2021). However, this method usually produces over-smoothed videos without realistic texture details (see Figure 1, I2VGen-XL), since the adopted bilinear upsampling cannot generate more spatial details. More importantly, the noising-denoising process (*i.e.,* starting from $t = 600$) will substantially change the original video content, which cannot always be acceptable in practical applications.

In short, current generative video enhancement methods face several challenges. First, sequentially applying temporal and spatial super-resolution is redundant, as they are independently trained, but using similar training datasets. Thus, such design is both sub-optimal and inefficient during the inference. Second, existing refinement methods struggle in balancing between video quality and fidelity to the original content. More importantly, they cannot perform effective super-resolution for increasing spatial and temporal details, which limits their practicality. Third, previous generative video enhancement methods lack the flexibility in dealing with different upscaling factors and refinement strengths for spatial or temporal super-resolution and video refinement.

To this end, we propose *VEnhancer*, a generative space-time enhancement method that supports both spatial and temporal super-resolution with flexible space and time up-sampling scales, as well as has the ability to remove visual artifacts and flickering with good maintenance of video content. It is built upon a pretrained and fixed generative video prior (Zhang et al., 2023b), which supplies the generative ability for video enhancement. To condition the video generation on low-frame-rate and low-resolution videos, we design Space-Time Controller (ST-Controller) for effective conditioning in both spatial and time dimensions. Furthermore, to handle different up-sampling scales and reduce artifacts or flickering with different degrees, we propose a *space-time data augmentation* algorithm to construct the training data. In particular, at the training stage, we sample different step sizes for skipping frames, downscaling factors, and noise levels to synthesize diversified condition videos. To ensure the proposed ST-Controller be aware of the associated data augmentation applied to each input video, we propose the *video-aware conditioning*. In particular, for key frame, the condition latent, the embeddings of the associated downscaling factor $s$ and noise level $\sigma$ by noise augmentation are incorporated into ST-Controller through video-aware conditioning.

With these designs, VEnhancer is a single end-to-end trainable network that can handle both spatial and temporal super-resolution, as well as video refinement. Moreover, it also supports arbitrary space and time up-sampling scales, and also supports flexible control on refinement strength and generative strength as user may prefer. Extensive experiments have demonstrated VEnhancer's ability in enhancing generated videos (see Figure 1). In these experiments, it outperforms state-of-the-art real-world and generative video super-resolution methods for spatial super-resolution only. In the space-time super-resolution, VEnhancer also surpasses state-of-the-art methods as well as cascaded spatial and temporal diffusion super-resolution models. At last, on the public video

generation benchmark, VBench (Huang et al., 2023), VEnhancer can significantly improve the overall performance of existing text-to-video algorithms.

Our contributions can be summarized as below:

1. We propose VEnhancer, a generative space-time enhancement method that can achieve generative spatial and temporal super-resolution for different upsampling factors, as well as controllable video refinement in one video diffusion model for the first time.

2. To achieve the unified generative space-time enhancement, we devise ST-Controller for effective multi-frame condition injection based on a pretrained and fixed generative video prior. Besides, space-time data augmentation and the associated video-aware conditioning are proposed for training ST-Controller in an end-to-end manner.

3. VEnhancer surpasses existng state-of-the-art video super-resolution methods and space-time super-resolution methods in enhancing generated videos. Also, it could improve the performance of open-source text-to-video methods on public video generation benchmark.

## 2 RELATED WORK

### 2.1 VIDEO GENERATION

Recently, there have been substantial efforts in training large-scale T2V (Wang et al., 2024; Ho et al., 2022; Guo et al., 2023; Chen et al., 2023c; Gupta et al., 2023; Blattmann et al., 2023b; Wang et al., 2023b;a) models on large scale datasets. Some works (Blattmann et al., 2023b; Wang et al., 2023b;a) inflate a pre-trained text-to-image (T2I) model by inserting temporal layers and fine-tuning them or all parameters on video data, or adopts a joint image-video training strategy. In order to achieve high-quality video generation, (Ho et al., 2022; Wang et al., 2023b; Blattmann et al., 2023b) adopts multi-stage pipelines. In particular, cascaded video diffusion models are designed: One T2V base model that is followed by one or more frame interpolation and video super-resolution models. VideoLDM (Blattmann et al., 2023b), LaVie (Wang et al., 2023b), and Upscale-A-Video (Zhou et al., 2023) all develop the video super-resolution model based on $4\times$ sd (StableDiffusion (Rombach et al., 2022)) upscaler, which has an additional downsampled image for conditioning the generation. One drawback of this base model is losing quite a lot generative ability compared with T2I base models. On the contrary, I2VGEN-XL follows SDXL (Podell et al., 2023) and uses noising-denoising process (Meng et al., 2021) to refine the generated artifacts. However, this strategy could improve stability but cannot increase the space-time resolution. VEnhancer is based on a generative video prior, and could address temporal/spatial super-resolution and refinement in a unified model.

### 2.2 VIDEO ENHANCEMENT

**Video Super-Resolution.** Video Super-Resolution (VSR) is proposed to enhance video quality by upsampling low-resolution (LR) frames into high-resolution (HR) ones. Traditional VSR approaches(Cao et al., 2021; Chan et al., 2021; 2022a; Isobe et al., 2020a;b;c; Liang et al., 2024; 2022; Wang et al., 2019; Xue et al., 2019) often rely on fixed degradation models to synthesize training data pairs, which leads to a noticeable performance drop in real-world scenarios. To bridge this gap, recent advances(Chan et al., 2022b; Xie et al., 2023) in VSR have embraced more diversified degradation models to better simulate real-world low-resolution videos. To achieve photo-realistic reconstruction, Upscale-A-Video(Zhou et al., 2023) integrates diffusion prior to produce detailed textures, upgrading VSR performance into next level. **Space-Time Super-Resolution.** Space Time Video Super-Resolution (STVSR) aims to simultaneously increase the resolutions of video frames in both spatial and temporal dimensions. Deep-learning based approaches(Haris et al., 2020; Kim et al., 2020; Xiang et al., 2020; Chen et al., 2022) have achieved remarkable results on STVSR. STARNet(Haris et al., 2020) increases the spatial resolution and frame rate by leveraging the mutual information between space and time. FISR(Kim et al., 2020) propose a joint framework with a multi-scale temporal loss to upscale the spatial-temporal resolution of videos. (Xiang et al., 2020) proposes a one-stage STVSR framework, which incorporates different sub-modules for LR frame features interpolation, temporal information aggregation and HR reconstruction. VideoINR(Chen et al., 2022) utilize the continuous video representation to achieve STVSR at arbitrary spatial resolution and frame rate. Although these methods obtain smooth and high-resolution output videos, but they fail in generating realistic texture details.

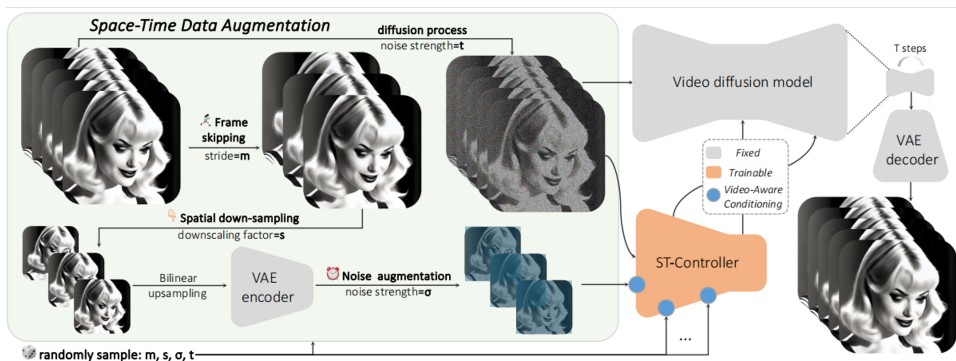

Figure 2: The overall framework. It consists of *space-time data augmentation* for constructing training data, and the associated *video-aware conditioning* for realizing diversified conditions across frames, as well as ST-Controller for multi-frame condition injection based on generative video prior.

## 3 PRELIMINARIES: VIDEO DIFFUSION MODELS

Our method is built on a pretrained video diffusion model (Zhang et al., 2023b), which is developed based on Stable Diffusion 2.1. Given an video $\mathbf{x} \in \mathbb{R}^{F \times H \times W \times 3}$, the encoder $\mathcal{E}$ first encodes it into latent representation $\mathbf{z} = \mathcal{E}(\mathbf{x})$ frame-by-frame, where $\mathbf{z} \in \mathbb{R}^{F \times H' \times W' \times C}$. Then, the forward diffusion and reverse denoising are conducted in the latent space. In the forward process, the noise is gradually added to the latent vector $\mathbf{z}$ in total $T$ steps. And for each time-step $t$, the diffusion process is formulated as follows:

$$\mathbf{z}_t = \alpha_t \mathbf{z} + \sigma_t \boldsymbol{\epsilon}, \tag{1}$$

where $\boldsymbol{\epsilon} \in \mathcal{N}(\mathbf{0}, \mathbf{I})$, and $\alpha_t$, $\sigma_t$ specify the noise schedule in which the corresponding log signal-to-noise-ratio ($\log[\alpha_t^2 / \sigma_t^2]$) decreases monotonically with $t$. And at time-step $T$, $q(\mathbf{z}_T) = \mathcal{N}(\mathbf{0}, \mathbf{I})$. As for backward pass, a diffusion model is used for iteratively denoising under the guidance of the text prompt $c_{text}$. By adopting v-prediction parameterization (Salimans & Ho, 2022), the U-Net denoiser $f_\theta$ learns to make predictions of $\mathbf{v}_t \equiv \alpha_t \boldsymbol{\epsilon} - \sigma_t \mathbf{z}$. The optimization objective is simply formulated as:

$$\mathcal{L}_{LDM} = \mathbb{E}_{\mathbf{z}, c_{text}, \boldsymbol{\epsilon} \sim \mathcal{N}(\mathbf{0}, \mathbf{I}), t} \left[ \| \mathbf{v} - f_\theta(\mathbf{z}_t, t, c_{text}) \|_2^2 \right]. \tag{2}$$

At the end, the generated videos are obtained through the VAE decoder: $\hat{\mathbf{x}} = \mathcal{D}(\mathbf{z})$.

## 4 METHODOLOGY

In this section, we introduce the main components of our method. The overall framework is illustrated in Figure 2. First, we present our architecture design in Section 4.1. Then we elaborate on the proposed *space-time data augmentation* in Section 4.2. In Section 4.3, we give a detailed description on our designed *video-aware conditioning*.

### 4.1 ARCHITECTURE DESIGN

The architecture is designed based on a pretrained video diffusion model. This video diffusion model is able to generate temporal-coherent content and high-quality texture details through iterative denoising. To upsample and refine a low-frame-rate and low-resolution videos in both spatial and temporal dimensions, the visual information should be incorporated into the video diffusion model carefully in order to obtain high-quality results with good fidelity to the input videos. High-quality generated videos stem from powerful generative models, while fidelity requires the algorithm to preserve the visual information of the input. Inspired by (Zhang et al., 2023a), we keep the pretrained video diffusion model untouched for preserving generative capability, and create a Space-Time Controller (ST-Controller) to obtain effective multi-frame condition injection for generative video enhancement. The architecture is illustrated in Figure 3.

The pretrained video diffusion model follows the design of stacking a sequence of interleaved spatial and temporal layers within the 3D-UNet (Blattmann et al., 2023b) architecture (gray blocks in Figure 3). Specifically, each spatial convolution layer (or attention layer) is followed by a temporal

convolution layer (or attention layer). The spatial layers are the same as those in Stable Diffusion 2.1, including ResBlocks (He et al., 2016), self-attention (Vaswani et al., 2017) layers, and cross-attention layers. The temporal convolution and attention layers are incorporated with their output layers initialized to zero and finetuned with video datasets. Specifically, the temporal convolution is one-dimensional convolution layer with a kernel size of 3, and the temporal attention is one-dimensional attention layer (Wang et al., 2023a). In this 3D-UNet, the video features that aligned by temporal layers in encoder will be skipped to the decoder, in which concatenation operation will be performed to combine skipped features with decoder features.

To build our proposed ST-Controller, we make a copy (both the architectures and weights) of the multi-frame encoder and middle block in 3D-UNet (orange blocks in Fig. 3) as the trainable condition network. This condition network takes low-frame-rate and low-resolution condition latents as well as full frames of noisy latents as inputs. Specifically, the condition latents and the associated augmentation parameters are incorporated into the condition network through our proposed video-aware conditioning. The output multi-scale temporal-coherent video features will be injected into the original 3D-UNet through newly added zero convolutions (yellow blocks in Fig. 3). The output features of the middle block in condition network will be added

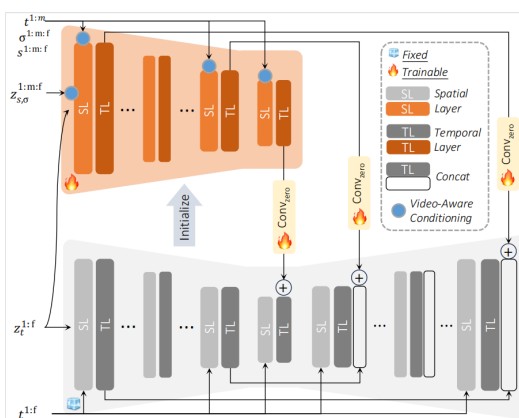

Figure 3: The architecture of VEnhancer.

back to the features of the middle block in 3D-UNet. While for output features of encoder blocks in condition network, their features will be added to the skipped video features in 3D-UNet, which are also produced by encoder blocks. The copied condition network, video-aware conditioning, and the newly added zero convolutions are trained simultaneously.

## 4.2 SPACE-TIME DATA AUGMENTATION

In this section, we discuss about how to achieve unified space-time super-resolution with arbitrary up-sampling space and time scales, as well as refinement with varying degrees. To this end, we propose a novel data augmentation strategy for both space and time axes. Details are discussed below.

***Time axis***. Given a sequence of high-frame-rate and high-resolution video frames $I^{1:f} = [I^1, I^2, ..., I^f]$ with frame length $f$, we use a sliding window across time axis to select frames. The frame sliding window size $m$ is randomly sampled from a predefined set, ranging from 1 to 8. This corresponds to time scales from $1\times$ to $8\times$. Note that $1\times$ time scale requires no frame interpolation, thus the multi-task problem downgrades to video super-resolution. After the frame skipping, we obtain a sequence of key frames $I^{1:m:f} = [I^1, I^{1+m}, I^{1+2\times m}, ..., I^f]$.

***Space axis***. Then, we perform spatial downsampling for these obtained key frames. Specifically, the downscaling factor $s$ is randomly sampled from $[1, 8]$, which represents $1\times \sim 8\times$ space super-resolution. When $s = 1$, there is no need to perform spatial super-resolution. All frames in one sequence are downsampled with the same downscaling factor $s$. Thus, we arrive at low-frame-rate and low-resolution video frames: $I^{1:m:f}_{\downarrow s}$. In practice, we should upsample them back to the original spatial sizes by bilinear interpolation before being passed to the networks, so we obtain $I^{1:m:f}_{\downarrow s, \uparrow s}$. Note that each space or time scale corresponds to different difficulty level, and thus the sampling is not uniform. Particularly, we set sampling probabilities of scales $4\times$ and $8\times$ based on a ratio of $1 : 2$, which is determined by their associated scale values.

Then, we use the encoder part of a pretrained variational autoencoder (VAE) $\mathcal{E}$ to project the input sequence to the latent space frame-wisely:

$$z^{1:m:f}_s = [\mathcal{E}(I^1_{\downarrow s, \uparrow s}), \mathcal{E}(I^{1+m}_{\downarrow s, \uparrow s}), \mathcal{E}(I^{1+2\times m}_{\downarrow s, \uparrow s}), ..., \mathcal{E}(I^f_{\downarrow s, \uparrow s})]. \quad (3)$$

***Noise augmentation in latent space***. At this stage, we conduct noise augmentation to noise the latent condition information in varying degrees in order to achieve controllable refinement. This noise

augmentation process is the same as the diffusion process equation 1 used in the video diffusion model. Specifically, the condition latent sequence is corrupted by:

$$z_{s,t'}^{1:m:f} = \alpha_{t'} z_s^{1:m:f} + \sigma_{t'}\epsilon^{1:m:f}, \tag{4}$$

where $\alpha_{t'}$, $\sigma_{t'}$ determine the signal-to-noise-ratio at time-step $t'$, and $t' \in \{1, ..., T'\}$. Note that the pretrained video diffusion model adopts 1,000 steps ($T = 1000$ in equation 1). While the noise augmentation only needs to corrupt the low-level information, $T'$ is set to 300 empirically. For more intuitive denotation, we use $\sigma$ instead of $t'$. Finally, we arrive at $z_{s,\sigma}^{1:m:f} = \mathcal{E}(I_{\downarrow s,\uparrow s})_\sigma^{1:m:f}$.

The whole process of space-time data augmentation is summarized as follows:

$$I^{1:f} \rightarrow I^{1:m:f} \rightarrow I_{\downarrow s}^{1:m:f} \rightarrow I_{\downarrow s,\uparrow s}^{1:m:f} \rightarrow \mathcal{E}(I_{\downarrow s,\uparrow s})^{1:m:f} \rightarrow \mathcal{E}(I_{\downarrow s,\uparrow s})_\sigma^{1:m:f}. \tag{5}$$

### 4.3 VIDEO-AWARE CONDITIONING

Besides data augmentation, the corresponding conditioning mechanism should also be designed in order to influence the model training and avoid averaging performance for different space or time scales and noise augmentation. In practice, the condition latent sequence $z_{s,\sigma}^{1:m:f}$, the corresponding downscaling factor $s$, and augmented noises $\sigma$ are all considered as for conditioning. Please refer to Figure. 4 for more intuitive demonstration.

Given the synthesized condition latent sequence $z_{s,\sigma}^{1:m:f}$, we use one convolution with zero-initialization –$\texttt{Conv}_{\texttt{zero}}$ for connecting it to the condition network. Specifically, we have:

$$f_{out}^{1:f} = \texttt{Conv}(z_t^{1:f}), \tag{6}$$

$$f_{out}^{1:m:f} = \texttt{Conv}(z_t^{1:m:f}) + \texttt{Conv}_{\texttt{zero}}(z_{s,\sigma}^{1:m:f}), \tag{7}$$

where $\texttt{Conv}$ is the first convolution in the condition network, $z_t^{1:f}$ and $z_t^{1:m:f}$ denote the full frames and key frames of noisy latents at timestep $t$, respectively. Note that $\texttt{Conv}$ and $\texttt{Conv}_{\texttt{zero}}$ share the same hyperparameter configuration (*i.e.,* kernel size, padding, et.al.), As it is shown, only key-frame features in condition network will be added with the condition features, while others remain unchanged. This strategy enables progressive condition injection as the weights of $\texttt{Conv}_{\texttt{zero}}$ grows from zero starting point.

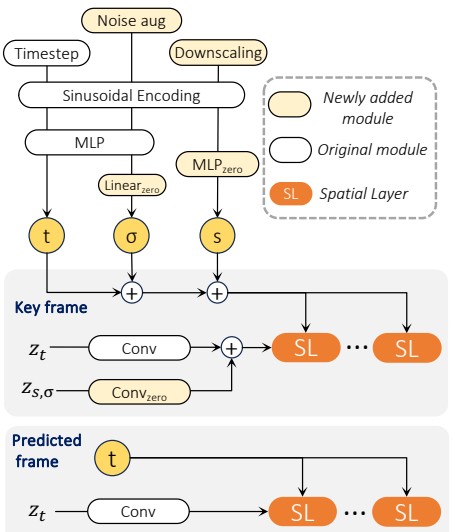

Figure 4: Video-aware conditioning. For frame that has condition image as input (key frame), we add the condition latent to the noisy latent after one convolution layer. Besides, the embeddings of noise level $\sigma$ and downscaling factor $s$ are added to the existing $t$ embedding, which will be broadcast to all spatial layers.

For conditioning regarding downscaling factor $s$ and noise augmentation $\sigma$, we incorporate them to the existing time embedding in the condition network. Specifically, for timestep $t$, sinusoidal encoding (Ho et al., 2020; Rombach et al., 2022; Vaswani et al., 2017) is used to provide the model with a positional encoding for time. Then, one $\texttt{MLP}$ (two linear layers with a SiLU (Elfwing et al., 2018) activation layer in between) is applied. Specifically, we have:

$$t_{emb} = \texttt{MLP}_t(\texttt{Sinusoidal}(t)), \quad t_{emb}^{1:f} = \texttt{Repeat}(t_{emb}, f), \tag{8}$$

where $t_{emb}^{1:f}$ is obtained by $\texttt{Repeat}$ $t_{emb}$ by $f$ times in the frame axis. This time embedding sequence will be broadcast to all ResBlocks in the condition network for timestep injection.

Also, we elucidate the conditioning for noise augmentation. As mentioned in equation 4, noise augmentation shares the same way as diffusion process, but with much smaller maximum timestep (*i.e.,* $T' = 300$). Thus, we reuse the encoding and mapping for timestep $t$ in diffusion process. After this, we add a linear layer with zero initialization (denoted as $\texttt{Linear}_{\texttt{zero}}$). To conclude, we have:

$$\sigma_{emb} = \texttt{Linear}_{\texttt{zero},\sigma}(\texttt{MLP}_t(\texttt{Sinusoidal}(\sigma))). \tag{9}$$

To achieve video-aware conditioning, we add $\sigma_{emb}$ only to the key frames. We repeat $\sigma_{emb}$ by $k$ times to obtain $\sigma_{emb}^{1:k}$, where $k$ is the number of key frames. The video-aware controlling is presented as follows:

$$t_{emb}^{1:m:f} = t_{emb}^{1:m:f} + \sigma_{emb}^{1:k}, \tag{10}$$

where the addition operation is performed frame-wisely.

Regarding downscaling factor $s$, the corresponding encoding, mapping and controlling are similar as above. In particular, we newly introduce one $\texttt{MLP}_{zero}$, in which the output layer is zero-initialized. The video-aware conditioning is performed as:

$$s_{emb} = \texttt{MLP}_{zero,s}(\texttt{Sinusoidal}(s)), \quad s_{emb}^{1:k} = \texttt{Repeat}(s_{emb}, k), \tag{11}$$

$$t_{emb}^{1:m:f} = t_{emb}^{1:m:f} + s_{emb}^{1:k}. \tag{12}$$

With our proposed *space-time data augmentation* and *video-aware conditioning*, VEnhancer can be well-trained in an end-to-end manner, and yields great performance in handling diversified conditions for generative enhancement. Here we provide a demonstration in Figure 5. For $4\times$ video super-resolution, we modify the input downscaling factor $s$ to produce different results. It is shown that more texture details are generated as $s$ grows (from *smooth* to *sharp*). This indicates that $s$ can determine how many details are generated through our proposed video-aware conditioning.

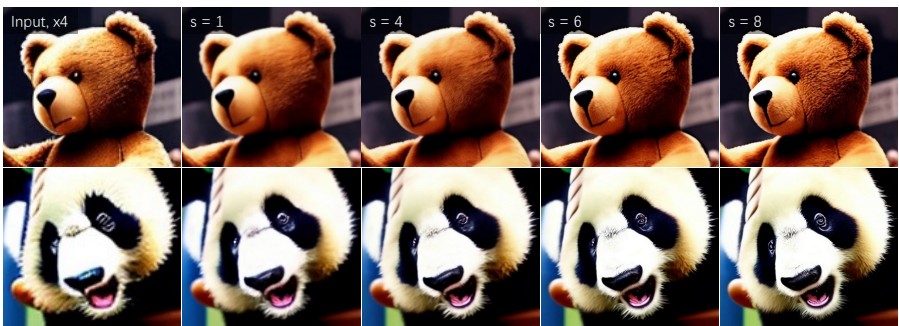

Figure 5: The effectiveness of video-aware conditioning. For video super-resolution ($4\times$), we modify the input downscaling factor from $s = 1$ to $s = 8$, and more texture details are generated. **Zoom in for best view.**

## 5    EXPERIMENTS

**Datasets.** We collect around 350k high-quality and high-resolution video clips from Panda-70M (Chen et al., 2024b) dataset and the Internet to constitute our training set. We train VEnhancer on resolution $1280 \times 720$ with center cropping, and the target FPS is fixed to $24$ by frame skipping. Regarding test dataset, we collect generated videos from state-of-the-art text-to-video methods. Practically, we select videos with large motions and diverse contents. This test dataset is denoted as AIGC2023, which is used to evaluate VEnhancer and baselines for video super-resolution and space-time super-resolution tasks. For evaluation on VBench, all generated videos based on the provided prompt suite are considered, resulting in around 5k videos.

**Implementation Details.** The batch size is set to 256. AdamW (Loshchilov & Hutter, 2017) is used as the optimizer, and the learning rate is set to $10^{-5}$. During training, we dropout the text prompt with a probability of 10%. The training process lasts about four days with 16 NVIDIA A100 GPUs. During inference, we use DPM-Solver (Lu et al., 2022) and perform 15 sampling steps with classifier-free guidance (*cfg*) (Ho & Salimans, 2022).

**Metrics.** Regarding evaluation for video super-resolution and space-time super-resolution on AIGC2023 test dataset, we use both image quality assessment (IQA) and video quality assessment (VQA) metrics. Specifically, MUSIQ (Ke et al., 2021) and DOVER (Wu et al., 2023a) are adopted. Moreover, we refer to video generation benchmark, VBench (Huang et al., 2023), for more comprehensive evaluation. Specifically, we choose **Dynamic Degree** (*i.e.,* whether it contains large motions), **Motion smoothness** (*i.e.,* how smooth the video is), and **Aesthetic Quality** for evaluation. Regarding evaluation for video generation, we consider all 16 evaluation dimensions from VBench.

## 5.1 COMPARISON WITH VIDEO SUPER-RESOLUTION METHODS

For video super-resolution, VEnhancer is compared with the state-of-the-art real-world video super-resolution method, RealBasicVSR (Chan et al., 2022b), and the state-of-the-art generative video super-resolution method, LaVie-SR (Wang et al., 2023b) (super-resolution). For more comprehensive comparison, we also include I2VGen-XL' (Zhang et al., 2023b) refinement model as our baseline.

Table 1: Quantitative comparison for video super-resolution ($4\times$) on AIGC2023 test dataset. **Red** and blue indicate the best and second best performance. The top 3 results are marked as gray .

| | DOVER↑ | MUSIQ↑ | Aesthetic Quality | Dynamic Degree | Motion Smoothness |
|---|---|---|---|---|---|
| LaVie-SR Wang et al. (2023b) | 0.8427 | 55.8428 | **0.6692** | 0.525 | 0.9710 |
| I2VGen-XL(refiner) Zhang et al. (2023b) | 0.5603 | 25.5988 | 0.6439 | 0.475 | **0.9835** |
| RealBasicVSR Chan et al. (2021) | 0.8252 | 50.5978 | 0.6622 | 0.550 | 0.9729 |
| Ours | **0.8586** | **59.4474** | 0.6671 | **0.550** | 0.9781 |

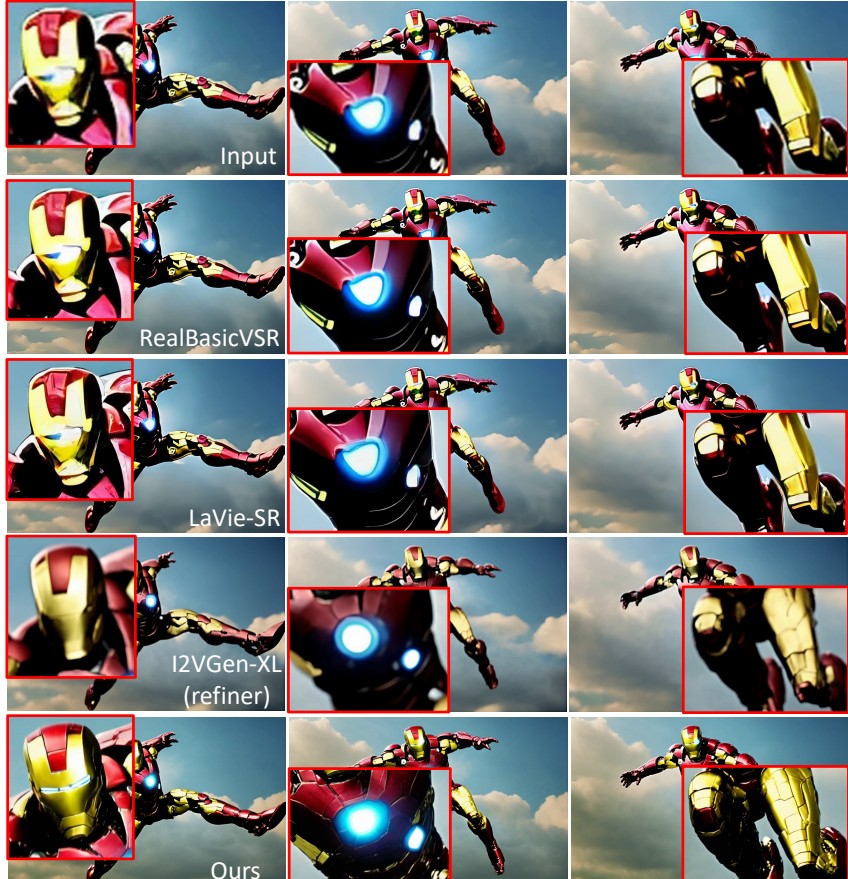

Figure 6: Visual comparison for video super-resolution ($4\times$) on AIGC2023 test dataset. Input resolution: $512 \times 312$; output resolution: $2048 \times 1280$. Prompt: *Iron Man flying in the sky*.

As shown in Table 1, VEnhancer outperforms both generative video super-resolution method (LaVie-SR), real-world video super-resolution method (RealBasicVSR), and generative video refinment method (I2VGen-XL, refiner) in most metrics, suggesting its outstanding enhancement ability for videos. Note that LaVie-SR surpasses RealBasicVSR in image/video quality (MUSIQ, DOVER, and **Aesthetic Quality**), as diffusion-based methods are better at generating sharp details. But LaVie-SR achieves the worst **Motion Smoothness**, indicating its insufficient capability in balancing video smoothness and video quality. I2VGen-XL's refiner could obtain the highest score in **Motion Smoothness**, but sacrifices the magnitude of motion significantly (worst **Dynamic Degree**). Moreover, it achieves very unsatisfactory results in metrics evaluating image and video quality. Because it uses bilinear interpolation for $\times 4$ upsampling, which produces very blurry results. Nevertheless, VEnhancer achieves overall best results with good balance among image/video quality, motion smoothness, and motion magnitude.

The visual comparison is presented in Figure. 6. The prompt is *"Iron man flying in the sky"*. The input video is already consistent with the prompt, but lacks details on the iron man suit. RealBasicVSR could remove some noises or artifacts of the generated videos as it incorporates complex degradation for model training. However, it fails in generating realistic details but produces over-smoothed results, since its generative ability is limited. On the other hand, the results of LaVie-SR contains more artifacts than input. Without successfully removing artifacts, the generative super-resolution model will enlarge the existing defects. The refiner of I2VGen-XL could achieve successful refinement but produces over-smoothed results. In contrast, VEnhancer could first remove unpleasing artifacts and refine the distorted content (*e.g.,* head region), and then generate faithfuls details (*e.g.,* helmet and armor) that are consistent with the text prompt.

## 5.2 COMPARISON WITH SPACE-TIME SUPER-RESOLUTION METHODS

For space-time super-resolution task, we compare two state-of-the-art space-time super-resolution methods: VideoINR (Chen et al., 2022) and Zooming-Slow-Mo (Xiang et al., 2020) (Zoom for short). We also consider LaVie's cascaded T-SR and S-SR DM-based pipeline: LaVie-FI (frame interpolation) + LaVie-SR (super-resolution) for more thorough comparison.

Table 2: Quantitative comparison for space-time super-resolution ($4\times$) on AIGC2023 test dataset. Red and blue indicate the best and second best performance. The top 3 results are marked as  gray .

| | DOVER↑ | MUSIQ↑ | Aesthetic Quality | Dynamic Degree | Motion Smoothness |
|---|---|---|---|---|---|
| LaVie-FI + LaVie-SR Wang et al. (2023b) | 0.8159 | **53.2128** | 0.6566 | 0.60 | 0.9857 |
| VideoINR Chen et al. (2022) | 0.7608 | 34.1060 | 0.6624 | 0.60 | 0.9933 |
| Zooming Slow-Mo Xiang et al. (2020) | 0.7328 | 33.8470 | 0.6624 | 0.55 | 0.9908 |
| Ours | **0.8609** | 51.2940 | **0.6710** | **0.60** | **0.9937** |

As shown in Table 2, we observe that our method surpasses all baselines in DOVER and **Aesthetic Quality**, showing its superior capability in generating sharp and realistic video content. Besides, it obtains highest scores in **Motion Smoothness** and **Dynamic Degree**, indicating VEnhancer's excellent ability in synthesizing stable temporal details. The cascaded T-SR and S-SR approach (LaVie-FI + LaVie-SR) obtains good scores in DOVER and MUSIQ, demonstrating DM-based methods' advantage in generation. However, its performance in temporal aspect is unsatisfactory due to its inferior capability in temporal refinement. We notice that state-of-the-art space-time super-resolution methods (VideoINR and Zooming Slow-Mo) behave well in **Motion Smoothness**. As both of them are optimized with reconstruction loss, the produced results are very smooth across frames. At a cost, they perform poorly in metrics regarding quality, such as DOVER and MUSIQ.

The visual comparison is illustrated in Figure. 7. The first and third columns present the low-resolution key frames. Note that the input frames are not consistent especially in the region of guitar strings. The cascaded T-SR and S-SR approach, LaVie-FI + LaVie-SR, can produce very sharp results for all frames (key and predicted ones). However, it generates messy contents which are not semantically aligned with prompt. Moreover, the generated details are changing across time, indicating severe flickering. For reconstruction-based methods (VideoINR and Zoom), the produced results are similar: lacking details and failing in improving the consistency of the original input frames. On the contrary, VEnhancer is not only able to achieve unified space-time super-resolution, but can also improve the temporal consistency of the generated videos by refinement (*i.e.,* guitar strings and raccoon hands).

## 5.3 EVALUATION ON IMPROVING VIDEO GENERATION

Here we evaluate VEnhancer's ability in improving state-of-the-art T2V methods. The baselines includes open-source T2V methods–VideoCrafter-2 (Chen et al., 2024a) (VC-2 for short), Lavie (Wang et al., 2023b), Open-Sora, CogVideoX (Yang et al., 2024), and professional video generation products–Pika and Gen-2. In particular, we enhance T2V results of CogVideoX-5B and VC-2.

The quantitative results are organized in Table 3. Before enhancement, CogVideoX-5B and VC-2 achieve the best and second best in *Semantic* compared with other baselines, demonstrating their superiority in generating video contents that are highly consistent to the VBench's prompt suite. Regarding *Quality*, they lag behind Pika. But with VEnhancer, VC-2 and CogVideoX-5B are able to achieve the highest and second highest scores in *Quality*. Besides, their scores regarding *Semantic* improves a lot, especially VC-2' (3.31% increase). This indicates that VEnhancer can improve the

Table 3: VBench Evaluation Results. This table compares the performance of open-source T2V methods and professional video generation products regarding two aspects (*Quality* and *Semantic*). A higher score indicates better performance. **Red** and blue indicate the best and second best performance. The top 3 results are marked as gray .

|  | LaVie
Wang et al. (2023b) | Open-Sora | Pika | Gen-2 | CogVideoX-5B
Yang et al. (2024) | VC-2
Chen et al. (2024a) | CogVideoX-5B
+Venhancer | VC-2
+VEnhancer |
|---|---|---|---|---|---|---|---|---|
| **Quality** | 78.78% | 80.71% | 82.92% | 82.47% | 82.75% | 82.20% | 82.99% | **83.28%** |
| **Semantic** | 70.31% | 73.30% | 71.77% | 73.03% | 77.04% | 73.42% | **77.52%** | 76.73% |
| **Overall** | 77.08% | 79.23% | 80.69% | 80.58% | 81.61% | 80.44% | 81.90% | **81.97%** |

semantic content and video quality at the same time, showing a powerful generative enhancement ability. More importantly, the advantage of adopting a two-stage pipeline is observed: the first T2V model focuses on generating semantic content and motions with good fidelity to the prompts, while the following enhancement model can improve the semantic in low-level and image quality, as well as temporal consistency. For visual results, **please see the video demonstration in supp.**

## 5.4 ABLATION STUDIES

**The Effectiveness of Noise Augmentation.** During training, the noise level regarding noise augmentation is randomly sampled within a predefined range. While during inference, one can change the noise level to achieve refinement with different strengths. In general, higher noise corresponds to stronger refinement and regeneration. We present the visual comparison among different noise levels in Figure 8. The first frame of one AI-generated video is presented in the left. It is of low-resolution and lacks details. Also, the original video has very obvious flickering. If we set $\sigma = 0$, VEnhancer will generate unpleasing noises in the background. As there is domain mismatch between the training data and testing data, the enhancement fails in handling unseen and challenging scenarios. Fortunately, we can mitigate this by adding noise in the condition latents for corrupting the noisy and unknown low-level details. As we increase the noise level, the artifacts are gradually vanishing. When $\sigma = 250$, the result is noise-clean, and has abundant semantic details.

**Arbitrary Up-sampling Scales for Spatial Super-Resolution.** Here we show that VEnhancer is able to up-sample videos with arbitrary scales. From Figure 9, we observe that VEnhancer could produce satisfactory results on different scales ($2.5\times$, $3\times$, $3.5\times$, $4\times$, and $4.5\times$), suggesting its flexibility and generalization in adapting to different tasks. In particular, given one frame of the generated video ($312 \times 512$), VEnhancer could improve the generated details when the up-sampling scale grows up. When $s = 2.5 \sim 3.5$, the panda's hand is less realistic. But it becomes better when $s = 4$ or $s = 4.5$. It is also noticed that the panda' fur is becoming more realistic as $s$ grows.

**Arbitrary Up-sampling Scales for Temporal Super-Resolution.** In this part, we show VEnhancer is able to achieve arbitrary up-sampling in time axis. Given two low-resolution key frames, we aim to up-sample them to high-resolution ones, and also interpolate several frames (ranging from 2 to 4) between them. As shown in Figure 10, the results are consistent across frames, showing not flicking or distortions. Besides, the spatial quality has also been significantly improved. As shown in the last row, $5\times$ frame interpolation yields smooth frames with generated contents: the shadow in the right leg is changing, showing a very natural transition. This indicates that diffusion-based frame interpolation has great capability in both motion and content generation.

## 6 CONCLUSION AND LIMITATION

In this work, we propose a generative space-time enhancement method – VEnhancer for video generation. It can achieve spatial super-resolution, temporal super-resolution and video refinement in one video diffusion model. We base on a pretrained generative video prior and build a Space-Time Controller (ST-Controller) for effective condition injection. Space-time data augmentation and video-aware conditioning are proposed to train ST-Controller in an end-to-end manner. Extensive experiments have demonstrated our superiority over state-of-the-art video super-resolution and space-time super-resolution methods in enhancing AI-generated videos. However, our work has several limitations. First, as it is based on diffusion models, the inference takes more time than one-step methods. Second, it may face challenges in handling AI-generated long videos, since the long-term (over 10s) consistency has not been addressed in this work.

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

# A APPENDIX

## A.1 COMPARISON WITH SPACE-TIME SUPER-RESOLUTION METHODS

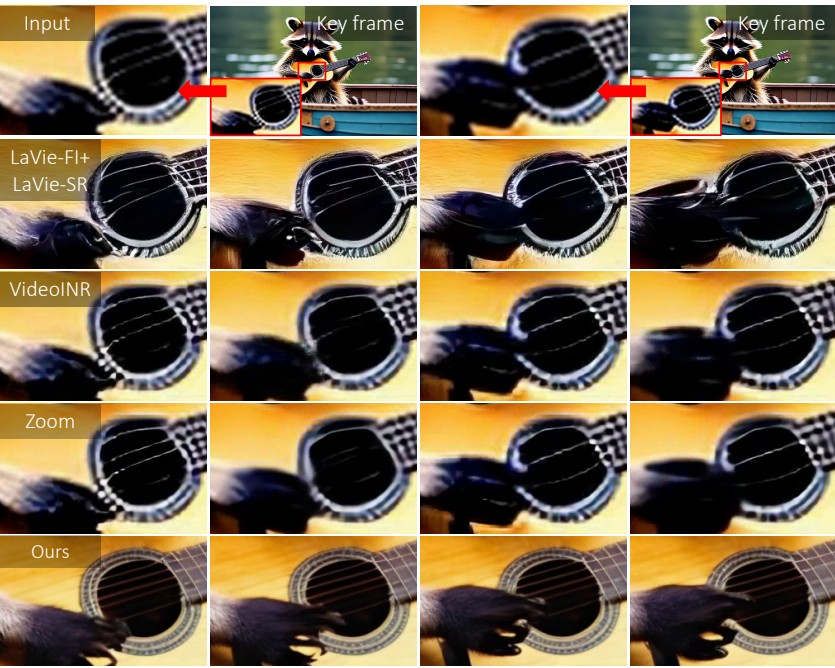

Figure 7: Visual comparison for space-time super-resolution on AIGC2023 test dataset. Prompt: *A cute raccoon playing guitar in a boat on the ocean.* **Zoom in for best view.**

## A.2 ABLATION STUDIES

**Comparison with Space-Time Super-Resolution Methods**

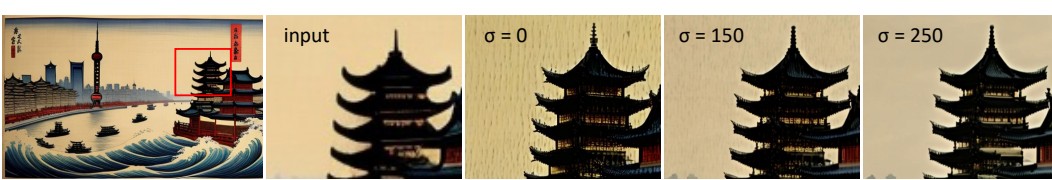

Figure 8: Visual comparison of setting different noise levels in noise augmentation during testing.

**Arbitrary Up-sampling Scales for Spatial Super-Resolution.**

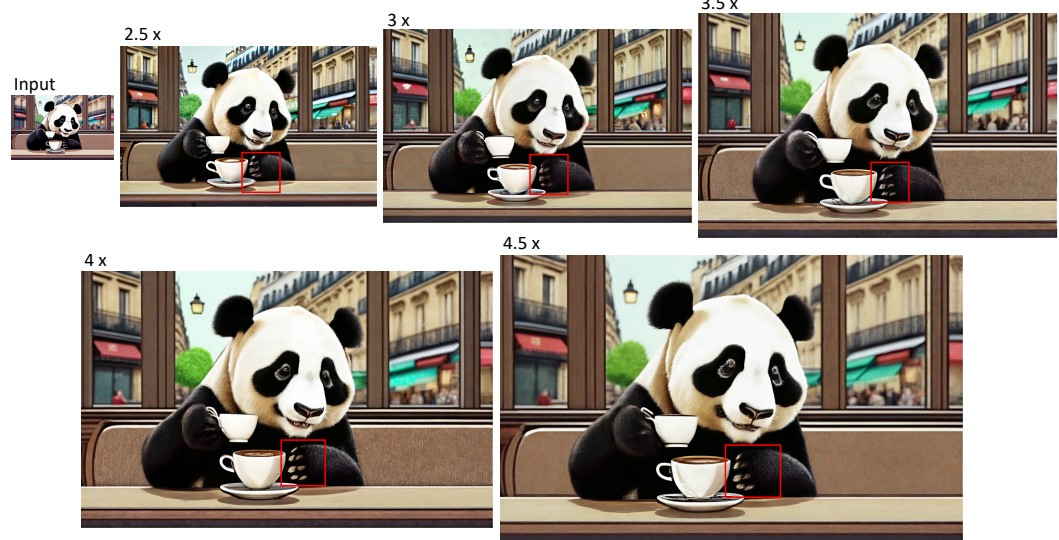

Figure 9: Visual results of different up-sampling scales (2.5×, 3×, 3.5×, 4×, and 4.5×) for spatial super-resolution during testing.

**Arbitrary Up-sampling Scales for Temporal Super-Resolution.**

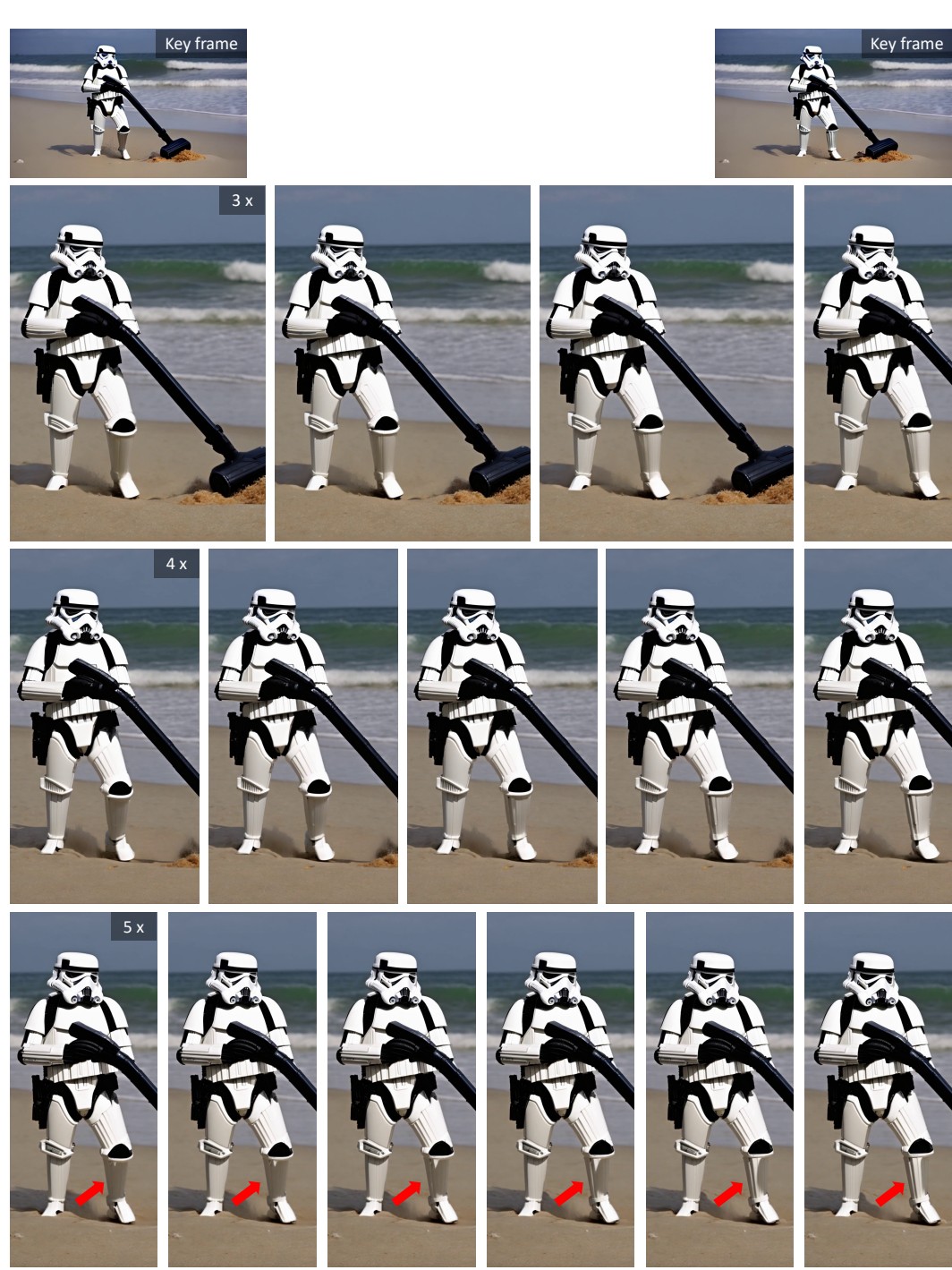

Figure 10: Visual results of different up-sampling scales ($3\times$, $4\times$, and $5\times$) for temporal super-resolution during testing.

